# Surface Properties and Cavitation Erosion Resistance of Cast Iron Subjected to Laser Cavitation Treatment

**Chunhui Luo [1],[\*],[†] and Jiayang Gu [2],[\*],[†]**

[1]  School of Mechanical Engineering, Changzhou Vocational Institute of Mechatronic Technology, Changzhou 213164, China
[2]  School of Intelligent Manufacturing, Jiangsu College of Engineering and Technology, Nantong 226001, China
[\*]  Correspondence: 18352861684@163.com (C.L.); jsgcxy_gjy@163.com (J.G.)
[†]  These authors contributed equally to this work.

**Abstract:** Laser cavitation is a novel surface modification technology using the impact of bubble collapse and laser-induced plasma to induce plastic deformation and produce compressive residual stress on material surfaces. The effects of laser cavitation on surface properties and the cavitation erosion resistance of cast iron were studied. In this work, three-dimensional morphology and residual stress distribution of the laser cavitation area under different laser parameters was obtained, the variation regularities of the topographic range and impact depth of the affected area was discussed, and the weight loss rate of cast iron under different defocusing amounts was studied. It was found that laser cavitation can effectively improve the anti-cavitation erosion property of the cast iron surface, and the optimal value was reached when the defocusing amount was H = 1 mm. Combined with the various defocusing amounts and the variation trend of the weight loss rate of cavitation erosion, the cavitation erosion time corresponding to each stage of the cast iron (incubation, rise, decay, and stability) was obtained.

**Keywords:** laser cavitation; surface properties; cavitation erosion resistance





## 1. Introduction

Cavitation is always the focus in the field of hydraulic machinery [1], as it leads to serious damages or erosion in hydraulic machinery and other components [2,3]. Sun, Ayli et al. [4,5] referred to the challenges of the sustainable development of hydropower. Therefore, studies regard the cavitation erosion resistance of material as not only a concern in the cavitation field but also an important factor in the fluid field [6,7]. Krella et al. [8] summarized the related properties of cavitation erosion resistance together with several methods for improving cavitation erosion resistance. In recent years, research regarding the cavitation erosion resistance of material has been divided into two parts, namely the cavitation erosion mechanism [9] and the corrosion resistance of metal material [10]. By establishing a cavitation theory model, the cavitation erosion mechanism has been studied to suppress or avoid cavitation. Reuter et al. [11] revealed an energy focusing mechanism during the non-spherical collapse of cavitation bubbles. Tong et al. [12] improved cavitation erosion resistance of AA5083 aluminum alloy by using laser shock peening. Furthermore, different reinforcement methods were used to improve the corrosion resistance of key components [13,14]. For the study of the cavitation erosion resistance of material, Jonda et al. [15] used deposition of the cermet coatings to protect the magnesium substrate, which showed low resistance to cavitation erosion and sliding wear when uncoated. Si et al. [16] used a novel surface strengthening method (ultrasonic shot peening) for improving the cavitation erosion resistance of 2024T351 Al alloy. Qin et al. [17] investigated the effect of compressive stress on the cavitation erosion–corrosion behavior of a nickel–aluminum bronze alloy, and the results showed that the alloy exhibited a selective phase corrosion of eutectoid "α + κiii".

Recently, scholars have discovered the positive effects of cavitation and have proposed the application of cavitation impacts to strengthen material [18,19]. Žagar et al. [20] investigated topographical behavior of precipitation-hardened magnesium alloy AZ80A subjected to cavitation peening, and found that cavitation peening also had a beneficial effect on compressive residual stresses. Soyama [21] applied cavitation strengthening technology to enhance the fatigue life of titanium alloy additive manufacturing. Gu et al. [22] used laser cavitation peening to impact mild steel, and the processing and strengthening mechanisms of material induced by laser cavitation was revealed. However, research on cavitation erosion resistance on material treated by laser cavitation processing has not been extensively discussed.

The purpose of this study is to illustrate the effect of different parameters of laser cavitation on surface properties and the cavitation erosion resistance of cast iron. Furthermore, a new method is promoted in this work to determine the degree of cavitation erosion resistance of the material by the diameter change rate of the treated area.

## 2. Experiments

### 2.1. Material Preparation

Cylindrical cast iron (HT200), with a diameter of $\Phi$16 mm and a height of 5 mm, was used as the test specimen. The tensile strength of HT200 cast iron was $\sigma_b \geq 200$ Mpa. The surface of the specimen was mechanically ground by SiC sandpaper with different grades of roughness (from 180# to 1500#), polished with 3.5 µm $SiO_2$ powder, and then cleaned with absolute ethanol. The chemical composition of the cast iron at room temperature is shown in Table 1.

**Table 1.** Chemical composition of HT200 cast iron (wt %).

| C | Cu | Si | Mo | Mn | Cr | P | S | Sb | Fe |
|---|---|---|---|---|---|---|---|---|---|
| 2.6~3.2 | 2.0~2.4 | 0.45~0.7 | 0.3 | 0.3 | 0.15 | 0.05 | 0.05 | 0.1 | Other |

### 2.2. Setup and Methods

The experimental setup of laser cavitation is schematically shown in Figure 1a. A Q-switched Nd:YAG laser generator (KSG1000, Kingder, Co., Ltd., Changzhou, China) with a wavelength of 1064 nm, a pulse width of 8 ns, an output energy between 200 mJ and 400 mJ, and a repetition frequency of 1 Hz was used to induce cavitation. The position of the focused laser beam was adjusted by using the optical arm with a length of 1.8 m. The laser beam diameter was 10 mm before entering the lens. The lenses were arranged by plano-concave, plano-convex, and plano-convex. The focal distance of the lenses were 20 mm, 100 mm, and 168 mm, respectively, and the spacing between the three lenses was 80 mm and 50 mm, respectively. The schematic representative regarding the beam path from air to water had a water height of 20 mm above the specimen. The standoff distance between the cavitation bubble center and the sample surface (H) was varied. Different laser energies were chosen to focus on the water, and the corresponding values were 200 mJ, 250 mJ, 300 mJ, 350 mJ, and 400 mJ. The sizes of the cavitation bubble generated during the laser cavitation were 2.0 mm, 2.8 mm, 3.3 mm, 3.6 mm, and 3.8 mm, depending on the laser energy. The different values of H employed during the cavitation experiments were 0 mm, 0.5 mm, 1.0 mm, 1.5 mm, and 2.0 mm, respectively, so the dimensionless parameters γ were 0, 0.36, 0.61, 0.83, and 1.05, respectively. There were 5 independent points on each specimen, corresponding to 5 different laser energies, in order to ensure the consistency of the standoff distance (H). The distribution of 5 points was 2 points on the left side and 3 points on the right side. The distance between the points was 3 mm, and each point acted only once. Each test was carried out at least 3 times in order to obtain consistent and comparable data.

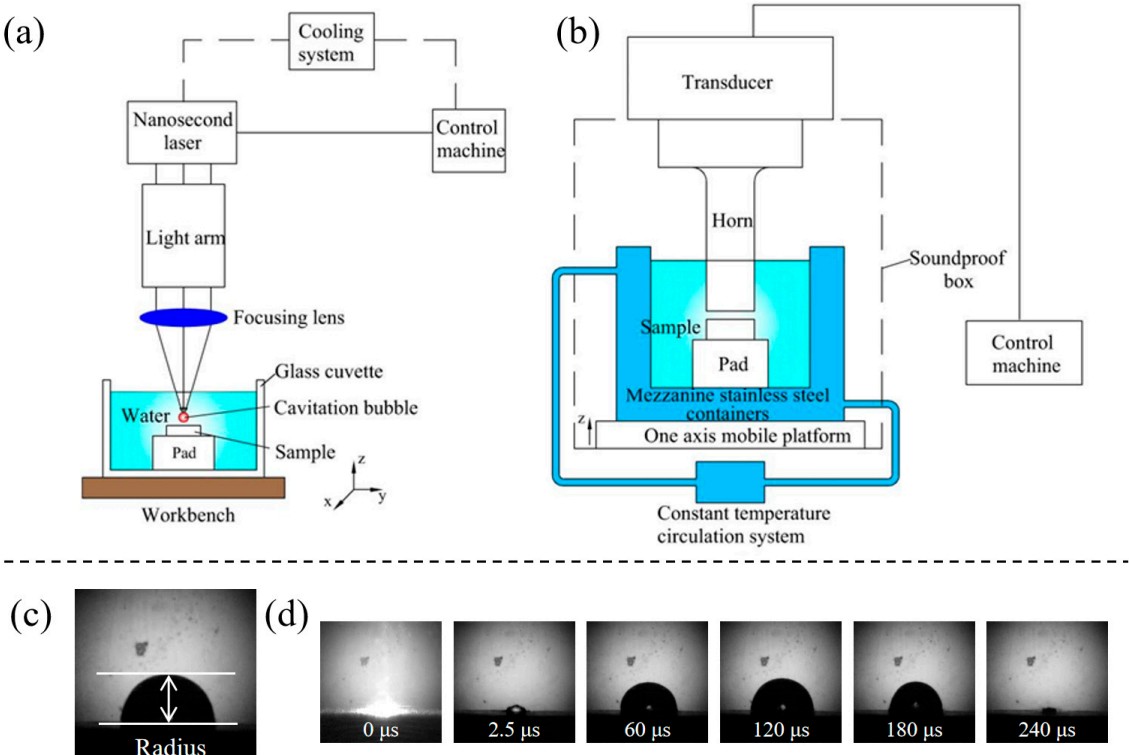

**Figure 1.** Experimental setups of (**a**) laser cavitation, (**b**) ultrasonic cavitation erosion system, (**c**) measurement of bubble size, and (**d**) sequence diagram of a cavitation bubble.

The schematic diagram of ultrasonic cavitation erosion is shown in Figure 1b. An ultrasonic signal generator (YJ98-IIIN) at 20 KHz frequency with a peak-to-peak vibratory amplitude of 60 μm and an ultrasonic power of 1200 W was used as an induced ultrasonic wave to provide energy for cavitation. Cast iron specimens were exposed to cavitation damage for periods of time ranging from 0 to 5 h. The specimen was placed at a stand-off distance of 1 mm below the tip. The water temperature was maintained through coolant circulation at 295 ± 1 K (22 ± 1 °C). The ultrasonic cavitation experiment was carried out in accordance with the standard of ASTM G32-98 (Standard Test Method for Cavitation Erosion Using Vibratory Apparatus, ASTM, Pennsylvania, PA, USA, 1998).

Figure 1c shows the measurement method of bubble size; the radius was measured according to high-speed photography, and the diameter was calculated from the value of the radius. The pulsating process of a cavitation bubble is shown in Figure 1d, including an expanding stage and a contracting stage.

### 2.3. Surface Properties Measurement

An ultradeep three-dimensional optical microscope (OLYMPUS-DSX500, Tokyo, Japan) was used to measure the three-dimensional topography of the specimen's surface. Residual stress was tested by an X350A residual stress tester (manufacturer is HDST) by X-ray diffraction with the sin2ψ method. The diameter of the X-ray beam was about 1 mm. The X-ray source was Cr–Ka ray, and the residual stress was calculated with a diffraction phase in the (220) plane. The feed angle of the ladder scanning was 0.1 deg/s. The scanning angle was from 125° to 133°. The X-rays were focused on a focal spot diameter of 1 mm and the residual stress obtained was the average value in the 1 mm range. For the measurement of the residual stress along the depth direction, the electro-polishing material removal method was used; the chemical was NaCl solution. The surface microhardness was measured by an HXD-1000TMSC/LCD Vickers microhardness tester with the function of image analysis. The loading pressure was 100 g, the holding duration was 15 s, and the standard deviation of measurement was about ±0.5%.

### 3. Results and Discussion

*3.1. Surface Integrity*

3.1.1. Surface Morphology

The three-dimensional morphologies of the treated specimens with laser irradiation energies of 200 mJ and 400 mJ at H = 0 mm, 1 mm, and 2 mm are shown in Figure 2, respectively. The cross-sectional profile of the corresponding three-dimensional surface of the specimens after laser cavitation is also shown in the figures.

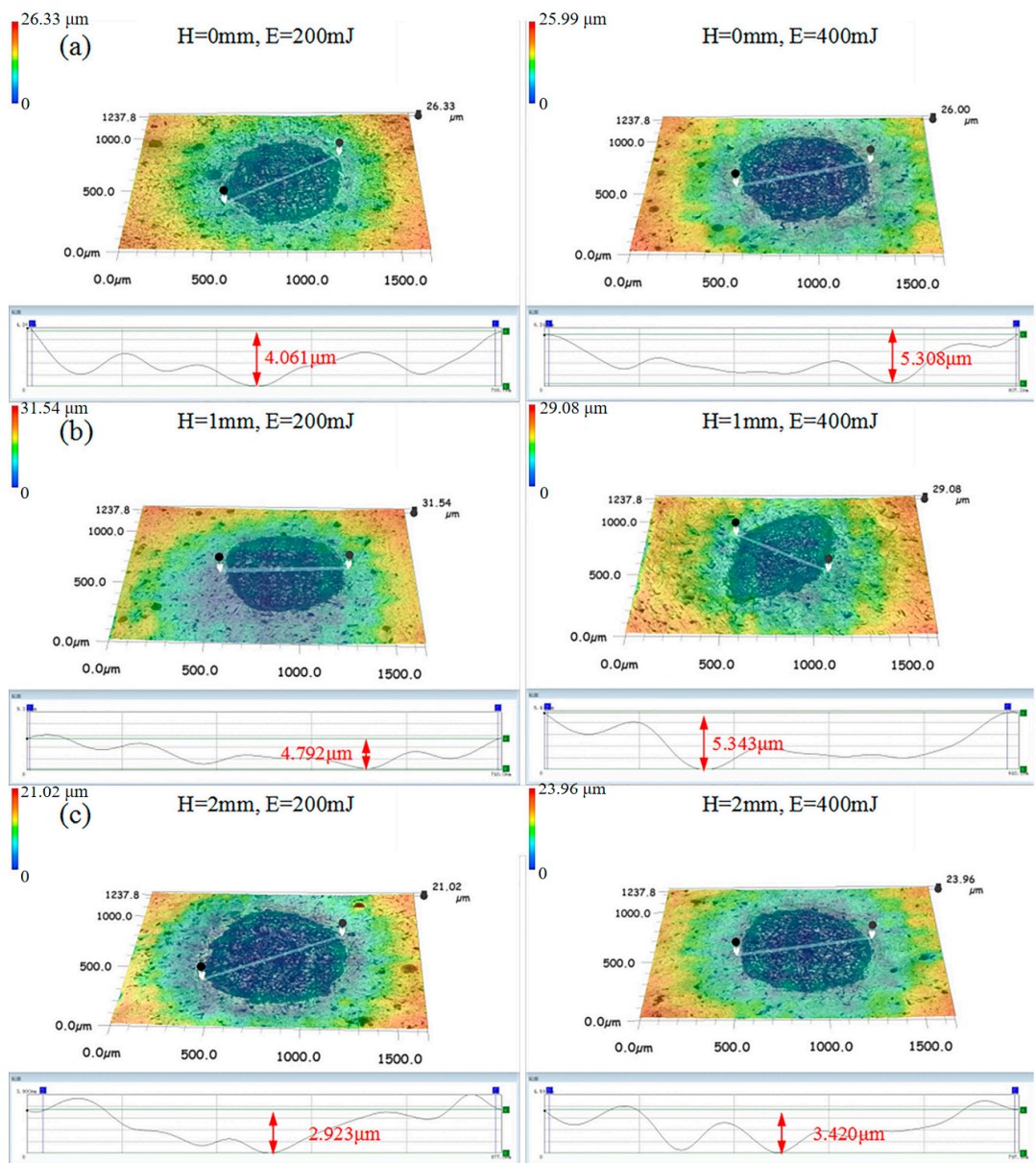

**Figure 2.** Three-dimensional morphologies of specimen treated by laser cavitation. (**a**) H = 0 mm, (**b**) H = 1 mm, and (**c**) H = 2 mm.

It shows in Figure 2a that the diameter of the treated area of a single laser beam on the specimen surface is approximately 500 μm. When the laser energy is 200 mJ, the maximum depth of the specimen surface is 4.061 μm. As the laser energy increases to 400 mJ, the maximum depth increases to 5.308 μm, indicating that the strengthening effect increases with an increase in laser energy. When H = 0 mm, the pits and microcracks in the treated area are obvious on the surface of the specimen; meanwhile, the damage of cavitation

increases significantly with the laser energy. It can be explained that when H = 0 mm, the laser focal point is located on the surface of the specimen. Since the laser energy does not have enough space to break through the liquid to form a bubble, a high-temperature and high-pressure plasma is formed on the specimen. The internal bubble pressure can reach 1700 MPa during the bubble collapse stage [23], which is far beyond the yield strength of cast iron, thus causing ablation.

Figure 2b shows the three-dimensional morphologies of the specimen surface after laser cavitation treatment when the laser energies are set to 200 mJ and 400 mJ at H = 1 mm. It is observed that both the cavitation treatment and the small amount of laser energy impact the sample surface, making the maximum pit depth increase from 4.792 μm to 5.343 μm. Moreover, the difference in pit depth decreases when H = 1 mm compared with when H = 0 mm. It can be explained that, when H = 1 mm, the laser beam has enough space to penetrate the aqueous medium to form a bubble. The indentation and permanent plastic deformation on the material surface is greatly improved by the cooperative impact effects of the plasma shock wave and cavitation jet when H =1 mm [24].

Similarly, as shown in Figure 2c, the maximum depth values of the sample pits at the two laser energies are only 2.923 μm and 3.420 μm when H = 2 mm, which is also less than the maximum pit depth when H = 0 mm and H = 1 mm. With the increase in the standoff distance, the intensity of the shock wave induced by the cavitation bubble rebound is weaker because the bubble energy is mostly consumed by the formation of the microjet. In addition, the shock wave and the cavitation jet compete against each other in this stage, and the main strengthening mechanism of the cavitation bubble on the specimen is the cavitation jet [25,26]. At the same time, the impact of laser ablation still exists but is weaker than H = 1 mm. So, the maximum depth value of the specimen gradually decreases when H = 2 mm.

Comprehensive analysis of the three-dimensional topography of the cast iron surface under different defocus amounts and laser energy parameters shows that, when the energy is in the range of 200 mJ to 400 mJ, with the further increase in energy, the direct laser impact and the shock wave and cavitation jet induced by the bubble collapse are more intense, and the maximum depth of the surface of the cast iron specimen also increases. When the laser energy is constant, the impact of the laser on the specimen goes through three stages as H increases. The three stages are divided into: (i) laser direct impact, (ii) shock wave of the cavitation and a smaller amount of laser impact, (iii) cavitation jet of the bubble and a much smaller amount of laser impact. The above results show that H has a significant effect on the surface morphology of the specimen. When the defocusing amount is too large, the energy and impact of laser and bubble collapse cannot be transferred to the specimen, and the maximum depth of the cast iron surface morphology finally shows a law of first increasing and then decreasing.

### 3.1.2. Residual Stress

Figure 3a shows the relationship between laser energy and residual stress. With the laser energy ranging from 200 mJ to 400 mJ, the residual compressive stress generates on the surface of the specimen after laser cavitation. Meanwhile, with the increase in laser energy, the residual compressive stress in the treated area gradually increases. The residual stress on the specimen surface is different when H is varied. When H = 0 mm, the pulse process is not complete because the cavitation bubble induced by the laser is too close to the specimen surface. With the increase in distance, the cavitation bubble has enough aqueous medium to complete the collapse process, the release shock wave, and the high-speed jet, which produce a larger compressive residual stress on the surface of the specimen. Therefore, the compressive residual stress distribution on the specimen surface is optimal when H = 1 mm. With a further increase in H to 2 mm, the impact strength of the jet and the intensity of the shock wave induced by the cavitation bubble rebound are weaker because the bubble energy is mostly consumed by water. At the same time, the effect of the laser ablation becomes weaker with the increase in distance during the

whole process, so the compressive residual stress begins to decrease gradually. The residual compressive stress can enhance the mechanical properties and prolong the fatigue life of the samples [27]. Therefore, within a certain range of laser energy, laser-induced cavitation has a strengthening effect on the specimens, and it increases with the increase in laser energy.

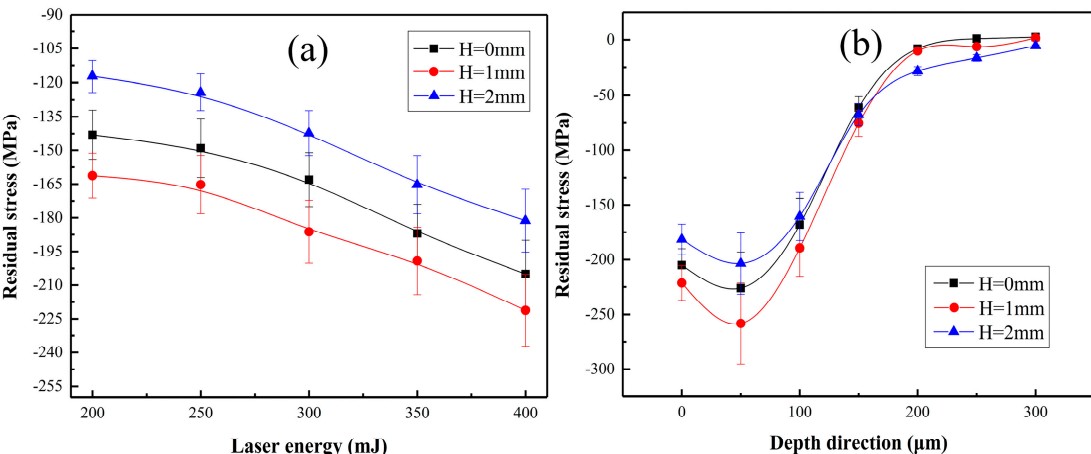

**Figure 3.** Residual stress diagrams for different defocusing amounts. (**a**) Relation between laser energy and residual stress; (**b**) residual stress in depth direction at 400 mJ.

As shown in Figure 3b, the in-depth residual stress distribution of the region treated with 400 mJ laser energy at different defocusing amounts is measured. The longitudinal residual stress distribution in the treated area is measured at intervals of 50 μm from the surface. It is observed that the highest value of the residual compressive stress in the treated area appears in the subsurface layer, which is significantly greater than the value of the residual compressive stress in the surface layer. This phenomenon can be explained by anti-deformation appearing on the surface of the specimens, which leads to the unloading effect of plastic deformation on the specimen surface, resulting in subsurface residual stress values greater than the specimen surface [28]. When H = 1 mm, the residual compressive stress at the subsurface layer at 50 μm in the treated region reaches −258 MPa.

In the case of H = 2 mm, when the depth exceeds 150 μm, the decrease in the residual compressive stress is slowed down and the value is relatively large compared with the other defocusing amount values, which indicates that laser cavitation at H = 2 mm has a greater effect on the depth of the specimens. The absorbed laser energy by liquid medium increases with the increase in the standoff distance. Under these circumstances, the cavitation jet plays a key role in strengthening the specimen [29]. With the accumulation of shock effects, the strain hardening can easily introduce plastic deformation and compressive residual stress on the target surface along the impact direction. When the depth is less than 150 μm, the maximum residual compressive stress appears at H = 1 mm, but when the depth exceeds 150 μm, the maximum residual compressive stress appears at H = 2 mm.

### 3.2. Cavitation Erosion Behavior

#### 3.2.1. Mass Loss

It is observed in Figure 4a that the mass loss of the specimen in the first 10 min of ultrasonic cavitation is higher but decreases gradually as the cavitation time increases. It can also be observed that the cumulative weight loss of the specimens after laser cavitation is lower than that without treatment. In the first 10 min, the weight loss of the specimen at H = 1 mm is 25.10 mg, compared with 58.34 mg for the untreated specimen. It can be seen that the surface of the polished specimen has poor cavitation erosion resistance. When H = 1.0 mm, the cumulative loss of the specimen is minimal, so the specimen has the best cavitation erosion resistance at this defocusing amount. When the distance between the laser focusing point and the surface of the material is too close, the laser cavitation cannot be completely carried out, and most of the laser energy will directly act on the surface of

the specimen in a short time, resulting in serious ablation, which is not conducive to the enhancement of the corrosion resistance of the material. When the distance between the laser focusing point and the surface of the material is too far, the effect of laser cavitation on the surface of the specimen will be greatly weakened by the liquid above the material. At this time, the corrosion resistance of the cast iron specimen cannot be effectively improved.

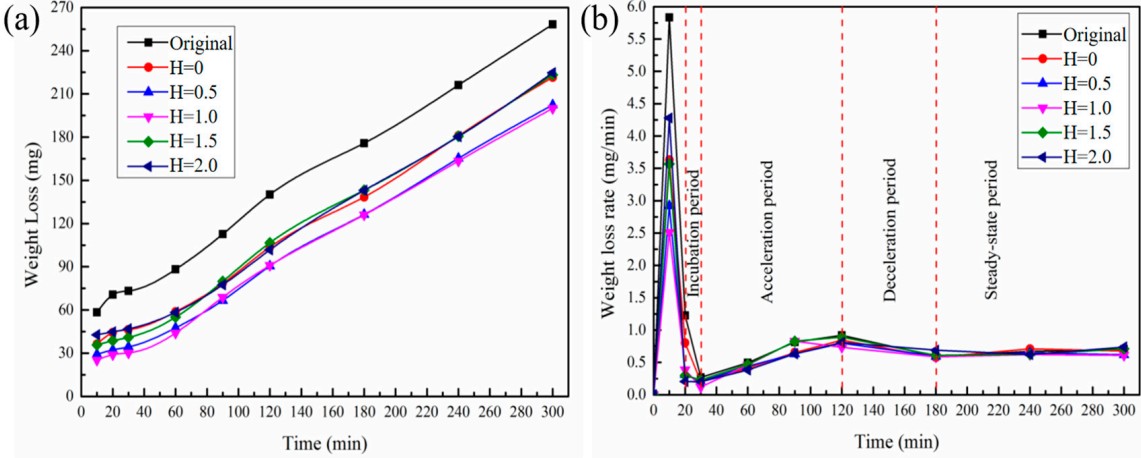

**Figure 4.** (**a**) Cumulative weight loss and (**b**) weight loss rate of cast iron in different defocusing amount.

Figure 4b shows the relationship between the weight loss rate of cavitation erosion and the time under different defocusing amounts. The process of cavitation weight loss of the specimens can be divided into four periods: incubation period, acceleration period, deceleration period, and steady–state period. After the cavitation erosion test for 20 min, the state of the specimen is close to that of the matrix. Therefore, the incubation period of the specimen is considered to be within 20–30 min. Between 30 and 120 min, the weight loss rate of the specimen gradually increases, that is, during the acceleration period of cavitation. When the ultrasonic cavitation erosion test is carried out between 120 and 180 min, the weight loss rate of the sample shows a decreasing trend, illustrating the deceleration period. In this period, the top surface of the specimen is completely removed under the repeated bubble collapse shock wave and jet impact stress, and the hardness of the subsurface material is greatly improved [30]. Since the cavitation erosion resistance is positively correlated with the hardness, the cavitation loss rate greatly decreases [31]. Subsequently, the cavitation weight loss rate of the specimen under different defocusing amounts is maintained at approximately 0.6 mg/min, and cavitation enters a steady–state period. In this period, the entire surface of the specimen is completely corroded and uniform crimp state morphology appears on the surface of the material. As the cavitation corrosion rate and the hardening rate of the material tend to balance, the mass loss rate remains essentially constant.

### 3.2.2. Erosion Morphology

Figure 5 shows the microscopic morphology of the cast iron surface without laser cavitation treatment after ultrasonic cavitation in water for 60 min, 180 min, and 300 min, respectively. It can be seen from Figure 5a,b that, after 60 min of cavitation erosion, the material still has a partly smooth surface, with areas not affected by the cavitation phenomenon. The destruction effect of the cavitation bubble collapse does not completely cover the entire surface of the cast iron. It is also possible that it is in the early stage of cavitation erosion ascent, and only the cavitation bubble with a large radius will cause a strong damage effect when it collapses on the surface of the material. However, there are obvious pits and cracks in most areas of the sample surface. These cracks originate from the junction between the graphite and the cast iron matrix, and gradually expand

from small cracks to large pits with the increase in cavitation erosion time until the surface of the specimen is damaged. Figure 5c–f show microstructures of the cast iron surface in the stable stage of cavitation erosion. Under the action of cavitation erosion, a large-scale plastic deformation occurs on the surface of the material, and the original crack propagates to the entire surface of the specimen. The surface roughness of the specimen at this stage is also much higher than in the rising stage of cavitation, which is due to the low melting point, specific heat capacity, and thermal conductivity of the gray cast iron. The energy and impact induced by the collapse of the cavitation bubble can easily lead to the plastic deformation of the surface of such materials, resulting in a large number of pits, protrusions, and other plastic accumulation phenomena, and finally a sponge-like cavitation morphology is formed. The material on the surface that has been damaged by cavitation adheres to the specimen and warps outward in a plate-like shape. At the same time, with the continuous action of cavitation, the curled and warped plate-like material is affected by the cavitation collapse shock wave and the microscopic surface. Under the impact of the cavitation jet, it finally falls off and exposes the underlying cast iron material, so that the cavitation weight loss rate of the cast iron specimen is in a roughly stable state.

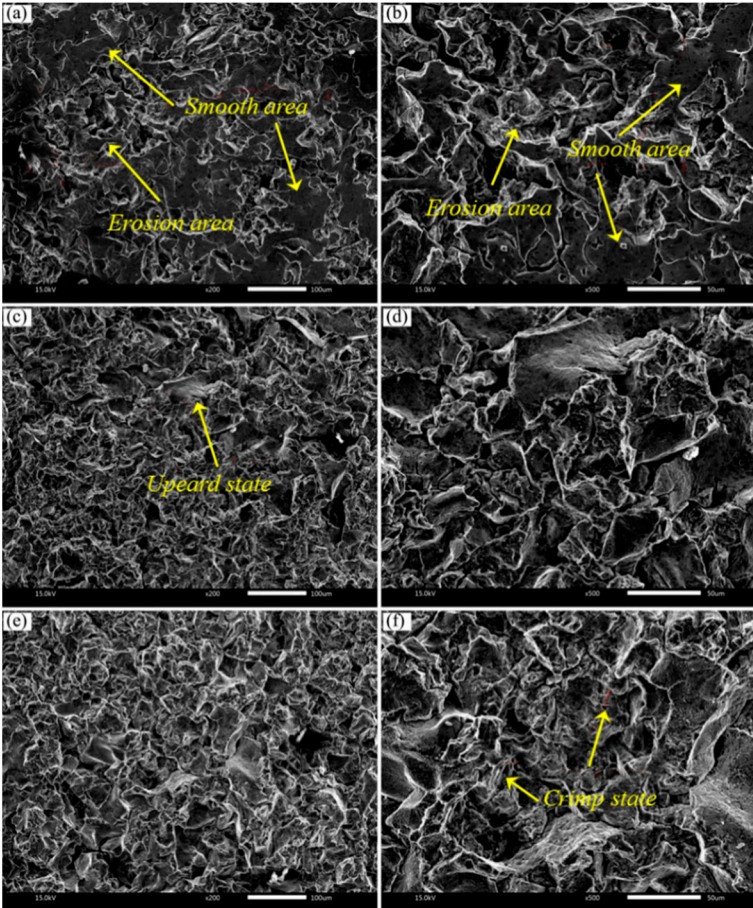

**Figure 5.** Surface morphologies of cast iron without laser cavitation after different cavitation erosion times of (**a**,**b**) 60 min, (**c**,**d**) 180 min, and (**e**,**f**) 300 min.

Figure 6 shows the surface morphologies of the specimens after different ultrasonic cavitation times for 200 mJ and 400 mJ laser energies at H = 0 mm. In Figure 6a–c, the sizes of the treated areas (circle diameters) are $\Phi$770.4 µm, $\Phi$695.92 µm, and $\Phi$648.24 µm, respectively. In Figure 6d–f, the sizes of the treated areas are $\Phi$807.2 µm, $\Phi$735.55 µm, and $\Phi$690.76 µm, respectively. With the increase in cavitation erosion time, the laser-affected areas gradually decrease from the periphery to the center. This indicates that when ultrasonic cavitation proceeds, cracks, pits, and other damages are more prone to appear

in the polished flat part of the specimens. The residual stress of the treated area presents significant improvement compared with the matrix. Therefore, the treated area (indicated by white dashed circles) can still be seen even after 120 min of ultrasonic cavitation, which indirectly means that the laser cavitation impact can effectively reduce the weight loss of the specimens. Moreover, the change in diameter of the specimen surface is analyzed by comparing different laser energies at the same cavitation time. The diameter differences between Figure 6a,b and Figure 6b,c at 200 mJ are 74.48 μm and 47.68 μm. Similarly, the diameter difference between Figure 6d,e and Figure 6e,f at 400 mJ is 71.65 μm and 44.79 μm. The higher laser energy range between 200 mJ and 400 mJ can increase the cavitation erosion resistance of the specimens.

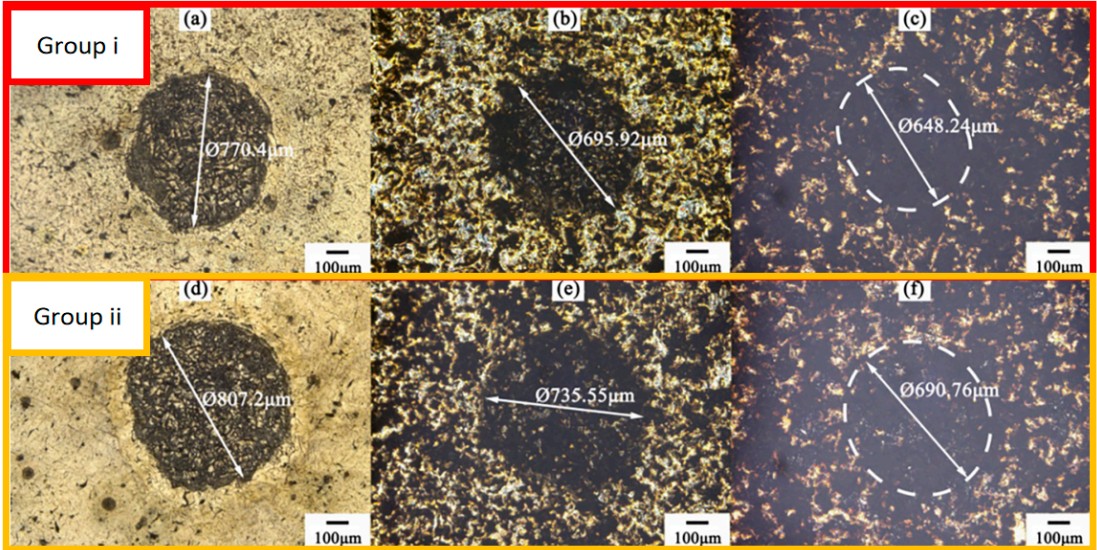

**Figure 6.** Surface morphologies of cast iron when H = 0 mm. (i) Laser energy of 200 mJ: (**a**) 0 min, (**b**) 60 min, and (**c**) 120 min; (ii) laser energy of 400 mJ: (**d**) 0 min, (**e**) 60 min, and (**f**) 120 min.

Figure 7 shows the surface morphology of the specimens at H = 1 mm. It can be seen that, with the increase in cavitation time, the treated area also decreases from the periphery to the center. When the laser energies are 200 mJ and 400 mJ, the diameter differences between Figure 7a,b and Figure 7b,c are 22.42 μm and 19.87 μm, respectively, at 200 mJ; and the diameter differences between Figure 7d,e and Figure 7e,f are 20.26 μm and 16.93 μm, respectively, at 400 mJ. At H = 1 mm, the diameter difference and the rate of change of each cavitation cycle are much smaller than those at H = 0 mm, which proves that the laser cavitation has a better favorable enhancement effect on the specimen when H = 1 mm. Moreover, the cavitation erosion resistance of the specimen is also better.

Figure 8 shows the surface morphologies of the specimens at H = 2 mm. The area change after the laser cavitation is consistent with the results of other defocusing amounts, both showing edge-to-center reduction. In addition, the difference in the treated areas for the 200 mJ energy are 87.96 μm and 39.84 μm from Figure 8a,b and Figure 8b,c and for 400 mJ are 71.51 μm and 33.86 μm from Figure 8d,e and Figure 8e,f. It is found that the rate of change of the diameter of the treated area at H = 2 mm is higher compared with H = 0 mm and 1 mm. The main reason is that, when the laser focal point is far away from the specimen surface, the laser energy is partly absorbed by the water, and the cavitation jet and the shock wave formed by the collapse of bubbles greatly reduce the effect on the specimens.

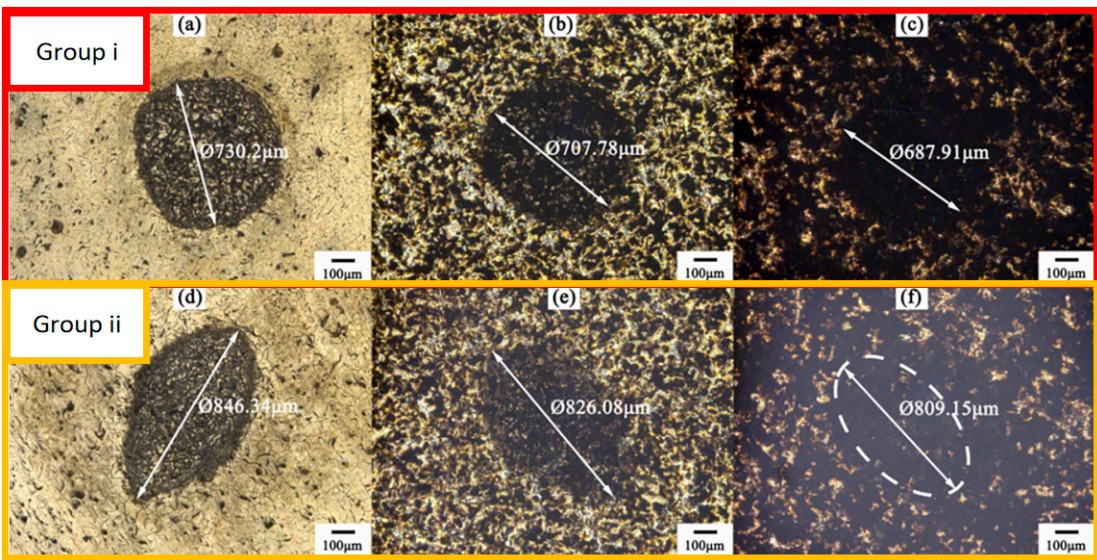

**Figure 7.** Surface morphologies of cast iron when H = 1 mm. (**i**) Laser energy is 200 mJ: (**a**) 0 min, (**b**) 60 min, and (**c**) 120 min; (**ii**) laser energy is 400 mJ: (**d**) 0 min, (**e**) 60 min, and (**f**) 120 min.

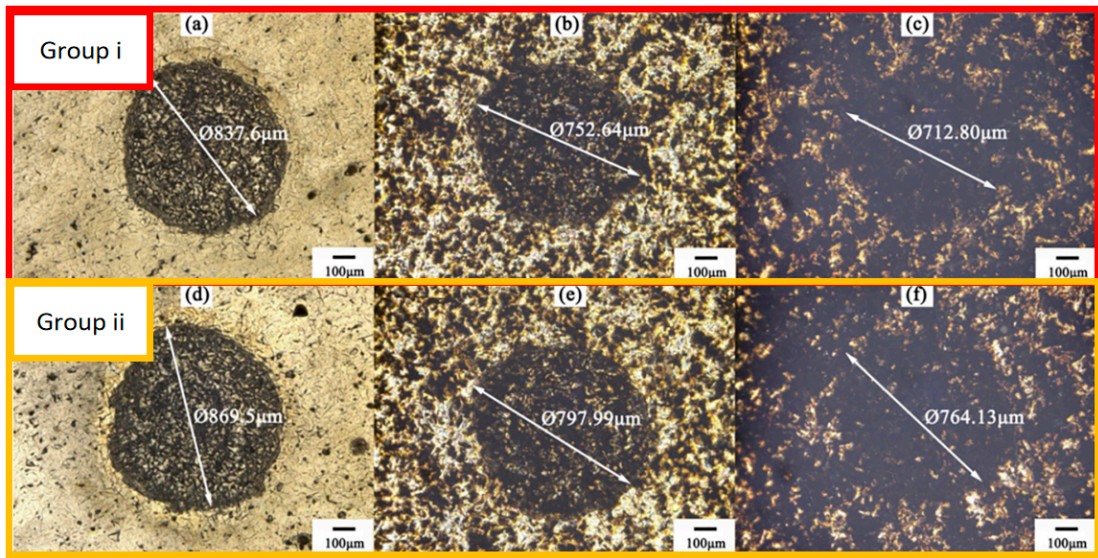

**Figure 8.** Surface morphologies of cast iron when H = 2 mm. (**i**) Laser energy is 200 mJ: (**a**) 0 min, (**b**) 60 min, and (**c**) 120 min; (**ii**) laser energy is 400 mJ: (**d**) 0 min, (**e**) 60 min, and (**f**) 120 min.

### 3.2.3. Hardness Changes after Cavitation Erosion

Figure 9 shows the trend of surface hardness of the specimens treated by various laser energies of 200 mJ, 300 mJ, and 400 mJ at H = 1 mm. As can be seen from the figure, the initial matrix hardness of the specimen is about 216 HV. When the ultrasonic cavitation is carried out for 30 min, the hardness of the specimen surface increases to 286 HV. The matrix hardness decreases and stabilizes at about 236 HV, when the cavitation time is further increased. When 200 mJ, 300 mJ, and 400 mJ laser energies are used for the laser cavitation, the initial hardness measurements of the treated areas are 315 HV, 328 HV, and 342 HV, respectively, which are far greater than the hardness of the cast iron matrix, and the hardness increases with the increase in laser energy. During the initial 20 min, the specimen is exposed to a large impact pressure due to the collapse of the cavitation bubbles; the continuous impact causes work hardening, and strain accumulation occurs in the impact zone. In addition, the formation of new dislocations leads to dislocation blocking and motion limitation, resulting in higher local hardness, so the hardness value of the specimen

increases initially. However, as time increases, the heat generated by the cavitation process, the repeated impact pressure, and the interaction between dislocations and grain structure may result in softening, that is, a decrease in hardness. In this stage, the hardness decreases and eventually stabilizes at 236 HV.

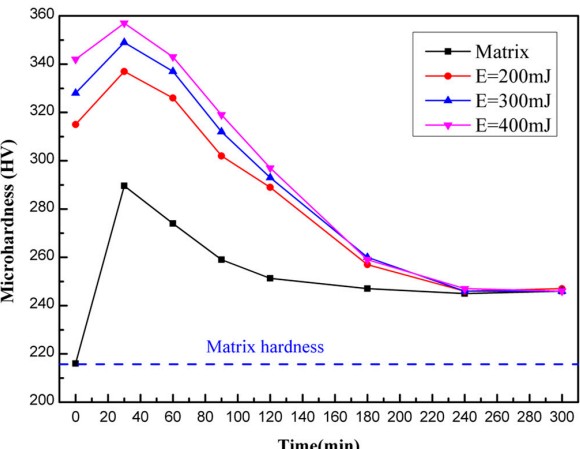

**Figure 9.** The change trend on surface hardness of cast iron with different laser energy when H =1 mm.

Meanwhile, comparing the hardness of the treated area with that of the substrate, it is found that the rate of hardness change in the treated area is smaller than that in the matrix during the cavitation incubation period, which is relatively higher when entering the acceleration and deceleration periods. This can be explained by the plastic deformation and work hardening of the specimens after laser cavitation are obviously weaker than that of the substrate. Therefore, the increase in hardness of the treated area is relatively small. The surface layer of the treated area is gradually peeled off by cavitation, exposing the substrate below the surface layer to the liquid, which results in a rapid decrease in the surface hardness value. When the ultrasonic cavitation is performed for 120 min, the impacted area of the specimen is completely destroyed. As the cavitation continues, the hardness of the exposed specimen decreases until the hardness tends to be consistent with the matrix hardness.

### 3.2.4. Cavitation Erosion Mechanism Analysis

Combined with the cavitation loss rate and the surface morphology of the cast iron samples before laser cavitation in different time periods, it can be found that, with the continuous ultrasonic cavitation, the microcracks begin to occur in the graphite sheet and the cast iron matrix area during the incubation stage. When the cavitation process is in the acceleration stage, the microcracks continuously expand outwards along the graphite sheet, resulting in pits and larger cracks. In the deceleration stage and stability stage of cavitation erosion, the plastic deformation of the material surface accumulates continuously, forming a kind of arch, curl, and concave-interlacing morphology. Finally, under the action of cavitation impact, the material surface outside the bulge is separated, making the material below continue to bear the effect of cavitation erosion. Therefore, the cavitation erosion mechanism of the cast iron matrix can also be understood as a process from microcrack initiation, to further crack propagation, to surface plastic accumulation, and finally to material surface peeling. In addition, surface roughness may affect the erosion rate to a certain extent. Large surface roughness means more pits on the surface, while the pits may expand due to the cavitation behaviors. As a result, the erosion rate increases with the surface roughness.

When cavitation erosion occurs in the region after laser cavitation, the cavitation mechanism is different from that of the cast iron matrix. After laser cavitation, the microstructure and casting defects of the graphite sheet existing on the surface of the specimen are improved with plastic deformation, and the microcrack initiation and propagation similar to the cavitation inoculation and rising stage of the cast iron matrix will not appear. With the

continuous transfer of cavitation energy to the surface of the specimen, the plastic deformation of the surface of the cast iron in the laser cavitation area will be gradually destroyed, and a small part of the material will be detached, but the deformation and destruction effect is weaker than that of the cast iron matrix, and the material in the sub-surface of the cast iron will also be gradually destroyed after a long time of cavitation erosion.

Mechanical properties such as residual compressive stress, hardness distribution, and surface morphology of metal materials are closely related to their cavitation resistance. With the increase in laser energy, the residual compressive stress appears in the laser cavitation region and reaches the peak at the subsurface layer. Compared with the cast iron without treatment, the surface hardness of cast iron after laser cavitation treatment is also greatly improved. Moreover, the energy generated by laser cavitation effectively forms micron plastic deformation on the surface of the specimen and covers the microstructure of the graphite sheet and the defects of cast iron on the surface of the specimen directly exposed to the liquid environment. Due to the micron-grade plastic deformation in the laser cavitation area, the surface finish will decrease accordingly, to a certain extent weakening the cavitation erosion resistance of the cast iron materials. However, based on the residual compressive stress and dramatic hardness increase, its mechanical performance is more significantly strengthened, so the cavitation erosion resistance on the surface of the cast iron material is further improved.

Compared with existing shot peening technology, the value of surface hardness and residual stress after laser cavitation treatment in this study is not exceptionally high. Soyama [32,33] listed the effects of several surface treatments. The effect of laser cavitation treatment is related to both laser parameters and liquid properties. Bubble size is a key factor for applied technologies related to cavitation. How to increase the size of cavitation bubbles is an important research trend in the field of cavitation applications.

Cavitation erosion resistance is a comprehensive concept, and the existing studies mainly characterize the cavitation erosion resistance through measuring the hardness, the residual stress, and the mass loss of the treated specimens [7,12]. However, the devices for detecting hardness and residual stress are mainly distributed in universities, research institutes, and testing companies. In addition, the detection steps are relatively complex, and the detection process may influence the integrality of specimens, such as measuring residual stress in depth direction.

Krella [8] proposed several other related parameters, such as ductility, fatigue strength, and fracture toughness. In this work, the diameter change rate of the cavitation affected area is considered as one of the parameters to represent the cavitation erosion resistance after cavitation treatment. The internal logic and the steps of this method are shown in Figure 10a,b. However, this is not a general method, which can only be employed on the materials after cavitation treatment.

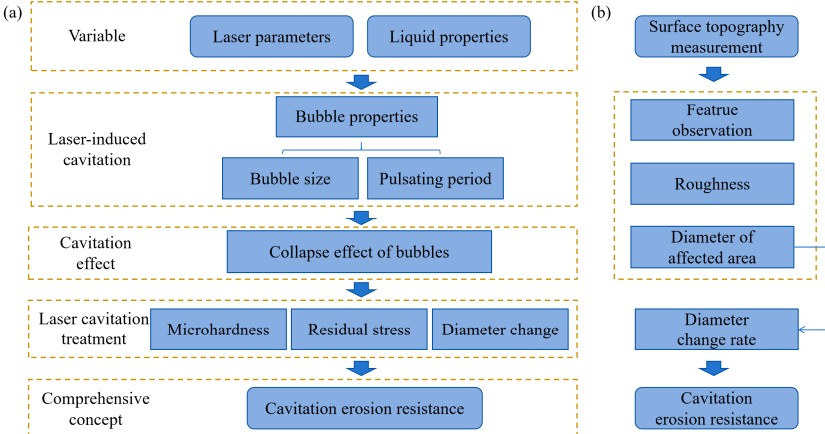

**Figure 10.** (**a**) The relationship of each variable and cavitation erosion resistance; (**b**) steps of the method proposed in this work.

## 4. Conclusions

These experimental studies were carried out to investigate the effect of laser cavitation on surface properties and the cavitation erosion resistance of cast iron. It is demonstrated that laser cavitation causes plastic deformation on the surface of the cast iron and introduces residual compressive stress. The degree of plastic deformation and residual compressive stress increases with the increase in laser energy, and the optimal effect is achieved when the defocusing amount is H = 1 mm. Laser cavitation may effectively improve the surface performance and cavitation corrosion resistance of the cast iron. With the increase in laser energy, the hardness of the laser cavitation area is significantly improved. Moreover, with the continuous cavitation, certain plastic deformations and work hardening phenomena on the surface of the specimen appear, and the hardness shows a trend of increasing first and then decreasing. During the increase in ultrasonic cavitation time, the cavitation erosion mechanism of cast iron changes from crack initiation and crack propagation to plastic deformation accumulation and surface material spalling. The reduction rate of the diameter of the treated area provides a new method for determining the level of cavitation erosion resistance after laser cavitation processing. However, more experiments should be performed on other common metal materials in fluids in order to analyze the effect of the laser cavitation technology on surface properties and cavitation erosion resistance more deeply.

**Author Contributions:** Writing—original draft, C.L.; Writing—review & editing, J.G. All authors have read and agreed to the published version of the manuscript.

**Funding:** The authors are grateful to the projects supported by the Postgraduate Research and Practice Innovation Program of Jiangsu Province (Grant No. KYCX21_3326).

**Data Availability Statement:** Data is contained within the article, and additional data sharing is not applicable to this article.

**Acknowledgments:** The authors are grateful to the support of Jiangsu University, Changzhou Vocational Institute of Mechatronic Technology and Jiangsu College of Engineering and Technology.

**Conflicts of Interest:** No potential conflict of interest was reported by the authors.

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
