# Peer review of "Surface Properties and Cavitation Erosion Resistance of Cast Iron Subjected to Laser Cavitation Treatment"

_metals, doi:10.3390/met13101793_

Round 1

Reviewer 1 Report

1. As cavitation erosion is affected by the surface roughness, please make a comment about the relation between surface roughness and erosion rate.

 2. “Different laser energies were chosen to focus in the water and the corresponding values were 200 mJ, 250 mJ, 300 mJ, 350 mJ and 400 mJ. The size of the cavitation bubble generated during laser cavitation were 2.0 mm, 2.8 mm, 3.3 mm, 3.6 mm, and 3.8 mm, depending on the laser energy.”

(1)  How to measure the bubble size?

(2)  Please show the evidence of laser cavitation such as high speed photography.

(3)  Is the produced bubble spherical bubble or hemi-spherical bubble?

Note that the impact induced by spherical bubble is weaker than that of hemi-spherical bubble.

(4)  In the reported reference (DOI:10.1016/j.jmatprotec.2019.01.030), pulsed laser of 350 mJ generated laser cavitation bubble 10 mm in diameter. In the paper, the pulsed laser of 350 mJ generated laser cavitation bubble 3.6 mm in diameter. It is too small, isn’t it? Note that the peening intensity is proportional to the volume of bubble (DOI:10.3390/app13116702).

3. In order to characterize the proposed laser cavitation treatment, it should be revealed the tendency of peening effect on work hardening and introduction of compressive residual stress comparing with the other peening method.

For example, Fig. 23 (Eq. (5)), Fig. 27 and Fig. 28 etc. of DOI:10.1016/j.jmatprotec.2022.117586

Author Response

Thanks for your comments and reminders, the manuscript has been optimized. In the revised manuscript, sentences with yellow background are the content revised according to the comments, and sentences with red characters mean grammar and spelling modification. In addition, Fig. 1, 2, 6, 7 and 8 have been optimized according to the comments.

Reviewer 2 Report

The paper presents an interesting study, related to the cavitation erosion resistance and the surface properties of a cast iron sample after a laser cavitation treatment.

The study can be added to a series of such experiments carried out by the authors in this direction, following different aspects related to the new type of surface treatment on different materials, in particular cast iron.

There are some concerns about the current experiment, namely:

- More details should be given about the 5 independent points, namely what these points associated with 5 different energies of the laser actually represent. I assume that it is about areas directed on the same sample used to estimate the effects, it being more efficient to change the orientation of the system than to replace the sample.

- Why were only 220 planes chosen for residual stress estimation? Is there any connection with the morphology/structure of the sample?

- Related to this aspect, in the previously published work (J. Gu, et al., Applied Surface Science 521 (2020) 146295), the chemical composition of the sample differs (less C, respectively Si).

Considering this variation in the chemical composition of different samples, the selection of the residual stress determination method should be linked to the structural characterization/examination performed on the studied sample.

Also, related to the selection of the 5 independent points mentioned previously, it seems that the 5 measurement points used in determining an average hardness have become individual points (the statistical correction ensured by the averaging of the results obtained at different points is thus eliminated).

The question is whether the result does not depend on the surface morphology from the point of measurement.

In figure 2 (H=1mm, E=400mJ), respectively figure 7 (ii), the geometry of the treated area (laser spot?) does not have the circular shape of the other cases. An explanation should be presented/offered in this regard.

On page 9, figure 5 is actually figure 6

A final aspect that I raise is the fact that the induction of residual stress can affect the mechanical properties. The direction offered in this study is strictly related to the surface properties. However, in real cases, will the proposed treatment affect the mechanical properties of the part?

Author Response

(The authors gave the same response as above.)

Reviewer 3 Report

Despite the manuscript reporting valuable results, it cannot be accepted in its present form. The manuscript must be written more clearly. In addition to spelling errors, there are many grammatical inconsistencies. For instance, the word "surface" is repeated twice in the first sentence of the Abstract.

Authors must address two technical concerns in a modified version of the manuscript. The authors must first explain the differences between the present work and the research reported in references [19] and [22]. The other technical issue arises from the following fact. The authors state that this work aims to illustrate a new methodology at the end of the Introduction section. However, the novelty of the proposed methodology is not discussed in the body of the manuscript. Instead, in the reviewer's opinion, the work is devoted to characterizing specific properties of the cast iron HT200 using a method already reported in the literature. Please clarify the technical contribution of the present work.

Besides correcting typos and grammatical errors, in a revised version of the manuscript, authors must pay attention to the following issues:

  • Extend the arguments regarding the relevance of references [1] to [14] for the present work.
  • Include units in Table 1
  • Report the brand and the main features of all the experimental devices employed in the research. 
  • Made readable the color scale in Figure 2. 
  • Correct the figure number in Figure 6. 

No further comments. 

Author Response

(The authors gave the same response as above.)

Round 2

Reviewer 2 Report

The revised version of the work meets the requirements, the authors clarifying the ambiguities present in the first version of the work through the answers given. I consider that the work can be accepted for publication in this form

Author Response

Responses to the reviewers:

  Thanks for your comments and reminders, the manuscript has been optimized. In the revised manuscript, sentences with yellow background are the content revised according to the comments, sentences with green background are the content revised according to the editorial office, and sentences with red characters mean grammar and spelling modification. In addition, a new figure has been added with several paragraphs.

Reviewer2

  1. The revised version of the work meets the requirements, the authors clarifying the ambiguities present in the first version of the work through the answers given. I consider that the work can be accepted for publication in this form

  Thanks very much for your attentions and comments.

Reviewer 3 Report

The modifications made to the original version of the manuscript significantly improved it. However, there are at least two aspects that must be addressed.

The first issue refers to the management of references. In the first nine lines of the Introduction section, 14 of the 33 references in the manuscript are cited, but they are not cited elsewhere. Please describe widely the relevance of the cited work concerning this manuscript.

The other issue is the purpose of the work. The authors declare that this manuscript proposes a new method to determine the degree of cavitation erosion resistance of the material by the diameter change rate of the treated area (lines 53-54). The authors must list the stages or steps of the proposed method textually. Please do not assume the reader can distinguish the method steps only by reading the manuscript.

Finally, please make and review the corrections carefully. There is a typo in line 375, a sentence already revised by the authors.  

Please make and review the corrections carefully.

Author Response

Responses to the reviewers:

  Thanks for your comments and reminders, the manuscript has been optimized. In the revised manuscript, sentences with yellow background are the content revised according to the comments, sentences with green background are the content revised according to the editorial office, and sentences with red characters mean grammar and spelling modification. In addition, a new figure has been added with several paragraphs.

Reviewer3

The modifications made to the original version of the manuscript significantly improved it. However, there are at least two aspects that must be addressed.

1.The first issue refers to the management of references. In the first nine lines of the Introduction section, 14 of the 33 references in the manuscript are cited, but they are not cited elsewhere. Please describe widely the relevance of the cited work concerning this manuscript.

  Thanks for your reminder, part of the references [1-14] have been specific explained in the Section ‘Introduction’, and some of them were cited in other paragraphs. Please refer to the sentences with yellow background for details.

  In addition, some of the references has been replaced to ensure that the recent 3 years' references are at least one-third of the whole, according to the comments of the editorial office. The sentences with green background are the related contents.

  1. The other issue is the purpose of the work. The authors declare that this manuscript proposes a new method to determine the degree of cavitation erosion resistance of the material by the diameter change rate of the treated area (lines 53-54). The authors must list the stages or steps of the proposed method textually. Please do not assume the reader can distinguish the method steps only by reading the manuscript.

  Thanks for your reminder, related contents have been added into the revised manuscript, including a figure and several paragraphs.

  1. Finally, please make and review the corrections carefully. There is a typo in line 375, a sentence already revised by the authors.  

  Thanks for your reminder, a new round of the corrections was carried out.
